# Discrete Flows: Invertible Generative Models of Discrete Data

**Dustin Tran**[1]    **Keyon Vafa**[12*]    **Kumar Krishna Agrawal**[1†]    **Laurent Dinh**[1]    **Ben Poole**[1]
[1]Google Brain    [2]Columbia University

## Abstract

While normalizing flows have led to significant advances in modeling high-dimensional continuous distributions, their applicability to discrete distributions remains unknown. In this paper, we show that flows can in fact be extended to discrete events—and under a simple change-of-variables formula not requiring log-determinant-Jacobian computations. Discrete flows have numerous applications. We consider two flow architectures: discrete autoregressive flows that enable bidirectionality, allowing, for example, tokens in text to depend on both left-to-right and right-to-left contexts in an exact language model; and discrete bipartite flows that enable efficient non-autoregressive generation as in RealNVP. Empirically, we find that discrete autoregressive flows outperform autoregressive baselines on synthetic discrete distributions, an addition task, and Potts models; and bipartite flows can obtain competitive performance with autoregressive baselines on character-level language modeling for Penn Tree Bank and text8.

## 1   Introduction

There have been many recent advances in normalizing flows, a technique for constructing high-dimensional continuous distributions from invertible transformations of simple distributions (Rezende and Mohamed, 2015; Tabak and Turner, 2013; Rippel and Adams, 2013). Applications for high-dimensional continuous distributions are widespread: these include latent variable models with expressive posterior approximations (Rezende and Mohamed, 2015; Ranganath et al., 2016; Kingma et al., 2016a), parallel image generation (Dinh et al., 2017; Kingma and Dhariwal, 2018), parallel speech synthesis (Oord et al., 2017; Ping et al., 2018; Prenger et al., 2018), and general-purpose density estimation (Papamakarios et al., 2017).

Normalizing flows are based on the change-of-variables formula, which derives a density given an invertible function applied to continuous events. There have not been analogous advances for discrete distributions, where flows are typically thought to not be applicable. Instead, most research for discrete data has focused on building either latent-variable models with approximate inference (Bowman et al., 2015), or increasingly sophisticated autoregressive models that assume a fixed ordering of the data (Bengio et al., 2003; Vaswani et al., 2017).

In this paper, we present an alternative for flexible modeling of discrete sequences by extending continuous normalizing flows to the discrete setting. We construct discrete flows with two architectures:

1. **Discrete autoregressive flows** enable multiple levels of autoregressivity. For example, one can design a bidirectional language model of text where each token depends on both left-to-right and right-to-left contexts while maintaining an exact likelihood and sampling.

2. **Discrete bipartite flows** enable flexible models with parallel generation by using coupling layers similar to RealNVP (Dinh et al., 2017) . For example, one can design nonautoregressive text models which maintain an exact likelihood for training and evaluation.

We evaluate discrete flows on a number of controlled problems: discretized mixture of Gaussians, full-rank discrete distributions, an addition task, and Potts models. In all settings, we find that stacking discrete autoregressive flows yields improved performance over autoregressive baselines, and that bipartite flows can reach similar performance to autoregressive baselines while being fast to generate. Finally, we scale up discrete bipartite flows to character-level language modeling where we reach 1.38 bits per character on Penn Tree Bank and 1.23 bits per character on text8 with generation speed over 100x faster than state-of-the-art autoregressive models.

## 1.1 Related Work

**Bidirectional models.** Classically, bidirectional language models such as log-linear models and Markov random fields have been pursued, but they require either approximate inference (Mnih and Teh, 2012; Jernite et al., 2015) or approximate sampling (Berglund et al., 2015). Unlike bidirectional models, autoregressive models must impose a specific ordering, and this has been shown to matter across natural language processing tasks (Vinyals et al., 2015; Ford et al., 2018; Xia et al., 2017). Bidirectionality such as in encoders have been shown to significantly improve results in neural machine translation (Britz et al., 2017). Most recently, BERT has shown bidirectional representations can significantly improve transfer tasks (Devlin et al., 2018). In this work, discrete autoregressive flows enable bidirectionality while maintaining the benefits of a (tractable) generative model.

**Nonautoregressive models.** There have been several advances for flexible modeling with nonautoregressive dependencies, mostly for continuous distributions (Dinh et al., 2014, 2017; Kingma and Dhariwal, 2018). For discrete distributions, Reed et al. (2017) and Stern et al. (2018) have considered retaining blockwise dependencies while factorizing the graphical model structure in order to simulate hierarchically. Gu et al. (2018) and Kaiser et al. (2018) apply latent variable models for fast translation, where the prior is autoregressive and the decoder is conditionally independent. Lee et al. (2018) adds an iterative refinement stage to initial parallel generations. Ziegler and Rush (2019) investigate latent variable models with continuous non-autoregressive normalizing flows as the prior. Aitchison et al. (2018) leverage fixed-point iterations for faster decoding of autoregressive models. In this work, discrete bipartite flows enable nonautoregressive generation while maintaining an exact density—analogous to RealNVP advances for image generation (Dinh et al., 2017). Most recently, Hoogeboom et al. (2019) proposed integer discrete flows, a concurrent work with similar ideas as discrete flows but with a flow transformation for ordinal data and applications to image compression and image generation. We find their results complement ours in illustrating the advantages of discrete invertible functions which do not require log determinant Jacobians.

## 2 Background

### 2.1 Normalizing Flows

Normalizing flows transform a probability distribution using an invertible function (Tabak and Turner, 2013; Rezende and Mohamed, 2015; Rippel and Adams, 2013). Let $\mathbf{x}$ be a $D$-dimensional continuous random variable whose density can be computed efficiently. Given an invertible function $f : \mathbb{R}^D \to \mathbb{R}^D$, the change-of-variables formula provides an explicit construction of the induced distribution on the function's output, $\mathbf{y} = f(\mathbf{x})$:

$$p(\mathbf{y}) = p(f^{-1}(\mathbf{y})) \det \left| \frac{d\mathbf{x}}{d\mathbf{y}} \right|. \tag{1}$$

The transformation $f$ is referred to as a flow and $p(\mathbf{x})$ is referred to as the base distribution. Composing multiple flows can induce further complex distributions that increase the expressivity of $p(\mathbf{y})$ (Rezende and Mohamed, 2015; Papamakarios et al., 2017).

### 2.2 Flow Transformation

For an arbitrary invertible $f$, computing the determinant of the Jacobian incurs an $O(D^3)$ complexity, which is infeasible for high-dimensional datasets. Thus, normalizing flows are designed so that the

determinant of the flow's Jacobian can be computed efficiently. Here, we review two popular flow transformations.

**Autoregressive flows.** Autoregressive functions such as recurrent neural networks and Transformers (Vaswani et al., 2017) have been shown to successfully model sequential data across many domains. Specifically, assume a base distribution $\mathbf{x} \sim p(\mathbf{x})$. With $\boldsymbol{\mu}$ and $\boldsymbol{\sigma}$ as autoregressive functions of $\mathbf{y}$, i.e. $\boldsymbol{\mu}_d, \boldsymbol{\sigma}_d = f(\mathbf{y}_1, \ldots, \mathbf{y}_{d-1})$, and $\boldsymbol{\sigma}_d > 0$ for all $d$, the flow computes a location-scale transform (Papamakarios et al., 2017; Kingma et al., 2016b),

$$\mathbf{y}_d = \boldsymbol{\mu}_d + \boldsymbol{\sigma}_d \cdot \mathbf{x}_d \qquad \text{for } d \text{ in } 1, \ldots, D.$$

The transformation is invertible and the inverse can be vectorized and computed in parallel:

$$\mathbf{x}_d = \boldsymbol{\sigma}_d^{-1}(\mathbf{y}_d - \boldsymbol{\mu}_d) \qquad \text{for } d \text{ in } 1, \ldots, D.$$

In addition to a fast-to-compute inverse, the autoregressive flow's Jacobian is lower-triangular, so its determinant is the product of the diagonal elements, $\prod_{d=1} \sigma_d$. This enables autoregressive flow models to have efficient log-probabilities for training and evaluation, but generation is sequential and inefficient.

**Bipartite flows.** Real-valued non-volume preserving (RealNVP) models uses another type of invertible transformation (Dinh et al., 2017) that nonlinearly transforms subsets of the input. For some $d < D$, a coupling layer follows a bipartite rather than autoregressive factorization:

$$\mathbf{y}_{1:d} = \mathbf{x}_{1:d} \tag{2}$$
$$\mathbf{y}_{d+1:D} = \boldsymbol{\mu}_{(d+1):D} + \boldsymbol{\sigma}_{(d+1):D} \cdot \mathbf{x}_{(d+1):D}, \tag{3}$$

where $\boldsymbol{\sigma}_{(d+1):D}$ and $\boldsymbol{\mu}_{(d+1):D}$ are functions of $\mathbf{x}_{1:d}$ with $\boldsymbol{\sigma}_{(d+1):D} > 0$ (we fix values for the lower indices, $\boldsymbol{\mu}_{1:d}$=0, $\boldsymbol{\sigma}_{1:d}$=1). Due to the bipartite nature of the transformation in coupling layers, we refer to them as bipartite flows. By changing the ordering of variables between each flow, the composition of bipartite flows can learn highly flexible distributions. By design, their Jacobian is lower-triangluar, with a determinant that is the product of diagonal elements, $\prod_{i=d+1}^{D} \sigma_i$.

Bipartite flows are not as expressive as autoregressive flows, as a subset of variables do not undergo a transformation. However, both their forward and inverse computations are fast to compute, making them suitable for generative modeling where fast generation is desired.

## 3 Discrete Flows

Normalizing flows depend on the change of variables formula (Equation 1) to compute the change in probability mass for the transformation. However, the change of variables formula applies only to continuous random variables. We extend normalizing flows to discrete events.

### 3.1 Discrete Change of Variables

Let $\mathbf{x}$ be a discrete random variable and $\mathbf{y} = f(\mathbf{x})$ where $f$ is some function of $\mathbf{x}$. The induced probability mass function of $\mathbf{y}$ is the sum over the pre-image of $f$:

$$p(\mathbf{y} = y) = \sum_{x \in f^{-1}(y)} p(\mathbf{x} = x),$$

where $f^{-1}(y)$ is the set of all elements such that $f(x) = y$. For an invertible function $f$, this simplifies to

$$p(\mathbf{y} = y) = p(\mathbf{x} = f^{-1}(y)). \tag{4}$$

This change of variables formula for discrete variables is similar to the continuous change of variables formula (Equation 1), but without the log-determinant-Jacobian. Intuitively, the log-determinant-Jacobian corrects for changes to the volume of a continuous space; volume does not exist for discrete distributions so there is no need to account for it. Computationally, Equation 4 is appealing as there are no restrictions on $f$ such as fast Jacobian computations in the continuous case, or tradeoffs in how the log-determinant-Jacobian influences the output density compared to the base density.

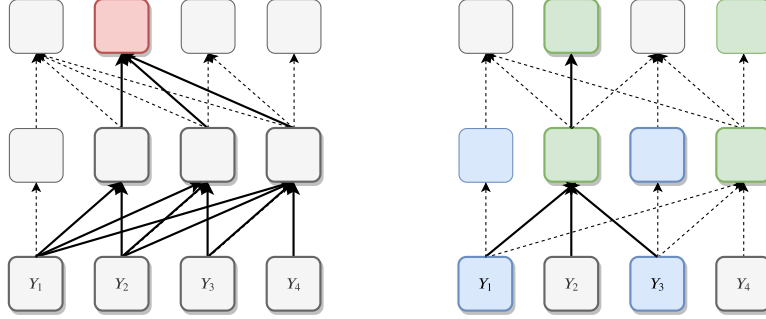

**Figure 1:** Flow transformation when computing log-likelihoods. **(a)** Discrete autoregressive flows stack multiple levels of autoregressivity. The receptive field of output unit 2 (red) includes left and right contexts. **(b)** Discrete bipartite flows apply a binary mask (blue and green) which determines the subset of variables to transform. With 2 flows, the receptive field of output unit 2 is $\mathbf{y}_{1:3}$.

### 3.2 Discrete Flow Transformation

Next we develop discrete invertible functions. To build intuition, first consider the binary case. Given a $D$-dimensional binary vector $\mathbf{x}$, one natural function applies the XOR bitwise operator,

$$\mathbf{y}_d = \boldsymbol{\mu}_d \oplus \mathbf{x}_d, \qquad \text{for } d \text{ in } 1, \ldots, D,$$

where $\boldsymbol{\mu}_d$ is a function of previous outputs, $\mathbf{y}_1, \ldots, \mathbf{y}_{d-1}$; $\oplus$ is the XOR function (0 if $\boldsymbol{\mu}_d$ and $\mathbf{x}_d$ are equal and 1 otherwise). The inverse is $\mathbf{x}_d = \boldsymbol{\mu}_d \oplus \mathbf{y}_d$. We provide an example next.

**Example.** Let $D = 2$ where $p(\mathbf{y})$ is defined by the following probability table:

|                  | $\mathbf{y}_2 = 0$ | $\mathbf{y}_2 = 1$ |
|------------------|--------------------|--------------------|
| $\mathbf{y}_1 = 0$ | 0.63               | 0.07               |
| $\mathbf{y}_1 = 1$ | 0.03               | 0.27               |

The data distribution cannot be captured by a factorized one $p(\mathbf{y}_1)p(\mathbf{y}_2)$. However, it can with a flow: set $f(\mathbf{x}_1, \mathbf{x}_2) = (\mathbf{x}_1, \mathbf{x}_1 \oplus \mathbf{x}_2)$; $p(\mathbf{x}_1)$ with probabilities $[0.7, 0.3]$; and $p(\mathbf{x}_2)$ with probabilities $[0.9, 0.1]$. The flow captures correlations that cannot be captured alone with the base. More broadly, discrete flows perform a multi-dimensional relabeling of the data such that it's easier to model with the base. This is analogous to continuous flows, which whiten the data such that it's easier to model with the base distribution (typically, a spherical Gaussian).

**Modulo location-scale transform.** To extend XOR to the categorical setting, consider a $D$-dimensional vector $\mathbf{x}$, each element of which takes on values in $0, 1, \ldots, K - 1$. One can perform location-scale transformations on the *modulo integer space*,

$$\mathbf{y}_d = (\boldsymbol{\mu}_d + \boldsymbol{\sigma}_d \cdot \mathbf{x}_d) \bmod K. \qquad (5)$$

Here, $\boldsymbol{\mu}_d$ and $\boldsymbol{\sigma}_d$ are autoregressive functions of $\mathbf{y}$ taking on values in $0, 1, \ldots, K-1$ and $1, \ldots, K-1$ respectively. For this transformation to be invertible, $\boldsymbol{\sigma}$ and $K$ must be coprime (an explicit solution for $\boldsymbol{\sigma}^{-1}$ is Euclid's algorithm). An easy way to ensure coprimality is to set $K$ to be prime; mask noninvertible $\boldsymbol{\sigma}$ values for a given $K$; or fix $\boldsymbol{\sigma} = 1$. Setting $K = 2$ and $\boldsymbol{\sigma} = 1$, it's easy to see that the modulo location-scale transform generalizes XOR. (We use $\boldsymbol{\sigma} = 1$ for all experiments except character-level language modeling.)

The idea also extends to the bipartite flow setting: the functions $(\boldsymbol{\mu}, \boldsymbol{\sigma})$ are set to $(0, 1)$ for a subset of the data dimensions, and are functions of that subset otherwise. Invertible discrete functions are widely used in random number generation, and could provide inspiration for alternatives to the location scale transformation for constructing flows (Salmon et al., 2011).

**Example.** Figure 2 illustrates an example of using flows to model correlated categorical data. Following Metz et al. (2016), the data is drawn from a mixture of Gaussians with 8 means evenly spaced around a circle of radius 2. The output variance is 0.01, with samples truncated to be between $-2.25$ and $2.25$, and we discretize at the 0.05 level, resulting in two categorical variables (one for $x$ and

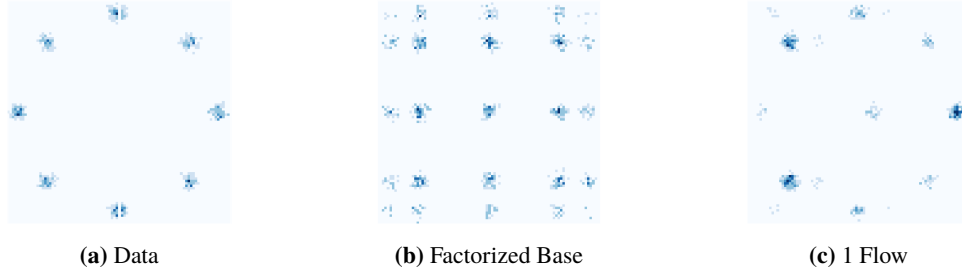

| (a) Data | (b) Factorized Base | (c) 1 Flow |

**Figure 2:** Learning a discretized mixture of Gaussians with maximum likelihood. Discrete flows help capture the modes, which a factorized distribution cannot. (Note because the data is 2-D, discrete autoregressive flows and discrete bipartite flows are equivalent.)

one for $y$) each with 90 states. A factorized base distribution cannot capture the data correlations, while a single discrete flow can. (Note the modulo location-scale transform does not make an ordinal assumption. We display ordinal data as an example only for visualization; other experiments use non-ordinal data.)

### 3.3 Training Discrete Flows

With discrete flow models, the maximum likelihood objective per datapoint is

$$\log p(\mathbf{y}) = \log p(f^{-1}(\mathbf{y})),$$

where the flow $f$ has free parameters according to its autoregressive or bipartite network, and the base distribution $p$ has free parameters as a factorized (or itself an autoregressive) distribution. Gradient descent with respect to base distribution parameters is straightforward. To perform gradient descent with respect to flow parameters, one must backpropagate through the discrete-output function $\boldsymbol{\mu}$ and $\boldsymbol{\sigma}$. We use the straight-through gradient estimator (Bengio et al., 2013). In particular, the (autoregressive or bipartite) network outputs two vectors of $K$ logits $\theta_d$ for each dimension $d$, one for the location and scale respectively. For the scale, we add a mask whose elements are negative infinity on non-invertible values such as 0. On the forward pass, we take the argmax of the logits, where for the location,

$$\boldsymbol{\mu}_d = \text{one\_hot}(\text{argmax}(\theta_d)). \tag{6}$$

Because the argmax operation is not differentiable, we replace Equation 6 on the backward pass with the softmax-temperature function:

$$\frac{d\boldsymbol{\mu}_d}{d\theta_d} \approx \frac{d}{d\theta_d}\text{softmax}\left(\frac{\theta_d}{\tau}\right).$$

As the temperature $\tau \to 0$, the softmax-temperature becomes close to the argmax and the bias of the gradient estimator disappears. However, when $\tau$ is too low, the gradients vanish, inhibiting the optimization. Work with the Gumbel-softmax distribution indicates that this approximation works well when the number of classes $K < 200$ (Maddison et al., 2016; Jang et al., 2017), which aligns with our experimental settings; we also fix $\tau = 0.1$.

**Limitations.** The extent to which the straight-through estimator works well can be unclear. The number of classes $K$ makes a difference, but there are also other factors. In particular, depth (number of flows) affects gradients, since the bias explicitly accumulates as each flow uses a gradient approximation. This is also the case for dimensionality (sequence length). We haven't found complexity of the networks parameterizing the flow to make a difference empirically, but this requires further investigation. As a reviewer mentioned, "true gradients" are also not well-defined, making gradient bias even less intuitive. Instead of examining gradient bias, one might formalize bias by comparing, for example, the minima obtained from optimizing the discrete loss to the minima obtained from optimizing the relaxed loss or moving to a setting where the parameters are stochastic. We leave better understanding of these limitations to future work.

|  | Autoregressive Base | Autoregressive Flow | Factorized Base | Bipartite Flow |
|---|---|---|---|---|
| $D = 2, K = 2$ | **0.9** | **0.9** | 1.3 | **1.0** |
| $D = 5, K = 5$ | 7.7 | **7.6** | 8.0 | **7.9** |
| $D = 5, K = 10$ | 10.7 | **10.3** | 11.5 | **10.7** |
| $D = 10, K = 5$ | 15.9 | **15.7** | 16.6 | **16.0** |

**Table 1:** Negative log-likelihoods for the full rank discrete distribution (lower is better). Autoregressive flows improve over its autoregressive base (bolded is best). Bipartite flows improve over its factorized base and achieve nats close to an autoregressive distribution while remaining parallel (bolded is best).

|  | AR Base | AR Flow |
|---|---|---|
| **number of states = 3** | | |
| $D = 9, J = 0.1$ | 9.27 | **9.124** |
| $D = 9, J = 0.5$ | 3.79 | **3.79** |
| $D = 16, J = 0.1$ | 16.66 | **11.23** |
| $D = 16, J = 0.5$ | 6.30 | **5.62** |
| **number of states = 4** | | |
| $D = 9, J = 0.1$ | 11.64 | **10.45** |
| $D = 9, J = 0.5$ | 5.87 | **5.56** |
| **number of states = 5** | | |
| $D = 9, J = 0.1$ | 13.58 | **10.25** |
| $D = 9, J = 0.5$ | 7.94 | **7.07** |

**Table 2:** Negative log-likelihoods on the square-lattice Potts model (lower is better). $D$ denotes dimensionality. Higher coupling strength $J$ corresponds to more spatial correlations.

## 4 Experiments

We perform a series of synthetic tasks to better understand discrete flows, and also perform character-level language modeling tasks. For all experiments with discrete autoregressive flows, we used an autoregressive Categorical base distribution where the first flow is applied in reverse ordering. (This setup lets us compare its advantage of bidirectionality to the baseline of an autoregressive base with 0 flows.) For all experiments with discrete bipartite flows, we used a factorized Categorical base distribution where the bipartite flows alternate masking of even and odd dimensions.

### 4.1 Full-rank Discrete Distribution

To better understand the expressivity of discrete flows, we examined how well they could fit random full-rank discrete distributions. In particular, we sample a true set of probabilities for all $D$ dimensions of $K$ classes according to a Dirichlet distribution of size $K^D - 1$, $\alpha = 1$. For the network for the autoregressive base distribution and location parameters of the flows, we used a Transformer with 64 hidden units. We used a composition of 1 flow for the autoregressive flow models, and 4 flows for the bipartite flow models.

Table 1 displays negative log-likelihoods (nats) of trained models over data simulated from this distribution. Across the data dimension $D$ and number of classes $K$, autoregressive flows gain several nats over the autoregressive base distribution, which has no flow on top. Bipartite flows improve over its factorized base and in fact obtain nats competitive with the autoregressive base while remaining fully parallel for generation.

### 4.2 Addition

Following Zaremba and Sutskever (2014), we examine an addition task: there are two input numbers with $D$ digits (each digit takes $K = 10$ values), and the output is their sum with $D$ digits (we remove

| | Test NLL (bpc) | Generation |
|---|---|---|
| 3-layer LSTM (Merity et al., 2018) | **1.18**[1] | 3.8 min |
| Ziegler and Rush (2019) (AF/SCF) | 1.46 | - |
| Ziegler and Rush (2019) (IAF/SCF) | 1.63 | - |
| Bipartite flow | 1.38 | **0.17 sec** |

**Table 3:** Character-level language modeling results on Penn Tree Bank.

the $D + 1^{th}$ digit if it appears). Addition naturally follows a right-to-left ordering: computing the leftmost digit requires carrying the remainder from the rightmost computations. Given an autoregressive base which poses a left-to-right ordering, we examine whether the bidirectionality that flows offer can adjust for wrong orderings. While the output is determnistic, the flexibility of discrete flows may enable more accurate outputs. We use an LSTM to encode both inputs, apply 0 or 1 flows on the output, and then apply an LSTM to parameterize the autoregressive base where its initial state is set to the concatenation of the two encodings. All LSTMs use 256 hidden units for $D = 10$, and 512 hidden units for $D = 20$.

For $D = 10$, an autoregressive base achieves **4.0** nats; an autoregressive flow achieves **0.2** nats (i.e., close to the true deterministic solution over all pairs of 10-digit numbers). A bipartite model with 1, 2, and 4 flows achieves **4.0**, **3.17**, and **2.58** nats respectively. For $D = 20$, an autoregressive base achieves **12.2** nats; an autoregressive flow achieves **4.8** nats. A bipartite model with 1, 2, 4, and 8 flows achieves **12.2**, **8.8**, **7.6**, and **5.08** nats respectively.

### 4.3 Potts Model

Given the bidirectional dependency enabled by discrete flows, we examined how they could be used for distilling undirected models with tractable energies but intractable sampling and likelihoods. We sampled from Potts models (the Categorical generalization of Ising models), which are a 2d Markov random field with pairwise interactions between neighbors (above/below, left/right, but not diagonally connected) (Wu, 1982). To generate data we ran 500 steps of Metropolis-Hastings, and evaluated the NLL of baselines and discrete flows as a function of the coupling strength, $J$. Low coupling strengths correspond to more independent states, while high coupling strengths result in more correlated states across space. For the base network, we used a single layer LSTM with 32 hidden units. For the flow network, we used an embedding layer which returns a trainable location parameter for every unique combination of inputs.

Table 2 displays negative log-likelihoods (nats) of trained models over data simulated from Potts models with varying lattice size and coupling strength. As Potts models are undirected models, the autoregressive base posits a poor inductive bias by fixing an ordering and sharing network parameters across the individual conditional distributions. Over data dimension $D$ and coupling $J$, autoregressive flows perform as well as, or improve upon, autoregressive base models. Appendix A includes samples from the model; they are visually indistinguishable from the data.

### 4.4 Character-Level Penn Tree Bank

We follow the setup of Ziegler and Rush (2019), which to the best of our knowledge is the only comparable work with nonautoregressive language modeling. We use Penn Tree Bank with minimal processing from Mikolov et al. (2012), consisting of roughly 5M characters and a vocabulary size of $K = 51$. We split the data into sentences and restrict to a max sequence length of 288. The LSTM baseline of Merity et al. (2018) uses 3 layers, truncated backpropagation with a sequence length of 200, embedding size of 400, and hidden size of 1850.[3] Ziegler and Rush (2019)'s nonautoregressive models have two variants, in which they use a specific prior with a conditionally independent likelihood and fully factorized variational approximation: AF/SCF uses a prior $\mathbf{z} \in \mathbb{R}^{T \times H}$ over latent time steps and hidden dimension that's autoregressive in $T$ and nonautoregressive in $H$; and IAF/SCF is

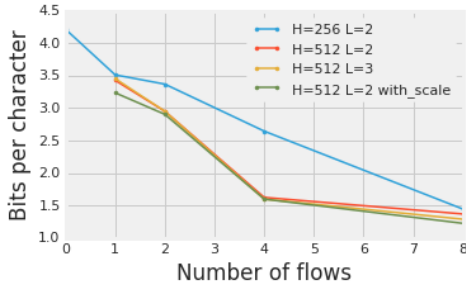

| | bpc | Gen. |
|---|---|---|
| LSTM (Cooijmans+2016) | 1.43 | 19.8s |
| 64-layer Transformer (Al-Rfou+2018) | **1.13** | 35.5s |
| Bipartite flow (4 flows, w/ $\sigma$) | 1.60 | **0.15s** |
| Bipartite flow (8 flows, w/o $\sigma$) | 1.29 | **0.16s** |
| Bipartite flow (8 flows, w/ $\sigma$) | 1.23 | **0.16s** |

**Figure 3:** Character-level language modeling results on text8. The test bits per character decreases as the number of flows increases. More hidden units $H$ and layers $L$ in the Transformer per flow, and applying a scale transformation instead of only location, also improves performance.

nonautoregressive in both $T$ and $H$. For the bipartite flow, we use 8 flows each with an embedding of size 400 and an LSTM with 915 hidden units.

Table 3 compares the test negative log-likelihood in bits per character as well as the time to generate a 288-dimensional sequence of tokens on a NVIDIA P100 GPU. The bipartite flow significantly outperforms Ziegler and Rush (2019), including their autoregressive/nonautoregressive hybrid. In addition, the generation time is over 1000x faster than the LSTM baseline. Intuitively, the use of bipartite flows means that we only have to perform one forward pass over the model as opposed to the 288 forward passes for a typical autoregressive model.

### 4.5 Character-Level text8

We also evaluated on text8, using the preprocessing of Mikolov et al. (2012); Zhang et al. (2016) with 100M characters and a vocabulary size of $K = 27$. We split the data into 90M characters for train, 5M characters for dev, and 5M characters for test. For discrete bipartite flows, we use a batch size of 128, sequence length of 256, a varying number of flows, and parameterize each flow with a Transformer with 2 or 3 layers, 512 hidden units, 2048 filter size, and 8 heads.

Figure 3 compares the test negative log-likelihood in bits per character as well as the time to generate one data point, i.e., a 256-dimensional sequence, on a NVIDIA P100 GPU. The bipartite flow reaches competitive performance, i.e., better than an LSTM baseline but not as good as the state-of-the-art bpc from the 235M parameter 64-layer Transformer (we're unfamiliar with previous nonautoregressive results to compare to). We also find that having a learned scale ("w/ $\sigma$") improves performance over fixing $\sigma = 1$ and only learning the location transform $\mu$. The bipartite flows' generation times are significantly faster than the baselines with upwards of a 100x speedup.

## 5 Discussion

We describe discrete flows, a class of invertible functions for flexible modeling of discrete data. Discrete autoregressive flows enable bidirectionality by stacking multiple levels of autoregressivity, each with varying order. Discrete bipartite flows enable nonautoregressive generation by flexibly modeling data with a sequence of bipartite-factorized flow transformations. Our experiments across a range of synthetic tasks and character-level text data show the promise of such approaches.

As future work, we're also investigating discrete inverse autoregressive flows, which enable flexible variational approximations for discrete latent variable models. An open question remains with scaling discrete flows to large numbers of classes: in particular, the straight-through gradient estimator works well for small numbers of classes such as for character-level language modeling, but it may not work for (sub)word-level modeling where the vocabulary size is greater than 5,000. In such settings, alternative gradient estimators or data representations may be fruitful. Lastly, there may be other invertible discrete functions used in cryptography and random number generation that could be leveraged for building up alternative forms for discrete flows (Salmon et al., 2011).

## Footnotes

*Work done as an intern at Google Brain. Supported by NSF grant DGE-1644869.

†Work done as an AI resident.

[3]The LSTM results are only approximately comparable as they do not apply the extra preprocessing step of removing sentences with >288 tokens.

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
