[Reviews · NeurIPS 2019]

Reviewer 1



------------------------------------------------------------------------------------------------------------------------------------------------------------------ POST REBUTTAL ------------------------------------------------------------------------------------------------------------------------------------------------------------------ The rebuttal has cleared most of my concerns and I am happy to maintain my score. ------------------------------------------------------------------------------------------------------------------------------------------------------------------ This paper ranks high in novelty as it is the first paper to consider discrete flows and also proposes the first discrete flow transformation layers (XOR and Additive). The experimental results are strong, especially on Text modelling. Moreover, the proposed method significantly computationally more efficient compared to competing approaches. The paper is very well written and easy to understand. However, the paper suffers from the following (few) shortcomings, 1. The capability of proposed XOR and Additive flow layers are unclear (even in 2D). E.g. in Figure 2 shows results only on a discretised Mixture of Gaussians. The fit to even this distribution is not perfect. The authors should consider a wider array of distributions to more convincingly demonstrate the capabilities of the proposed flow layers. E.g. discretised versions of the distributions analysed in [1]. 2. Some important details are unclear. E.g. what is the base distribution for sampling? Is it the factorised marginal distribution? If it is how is it estimated in high dimensions (given that the number of data samples needed for an accurate estimate would grow exponentially)? [1] Invertible Residual Networks, Behrmann et. al.

Reviewer 2



Originality: This paper is the first demonstration of flow-based models to discrete data. As such, the work is fairly novel. The flow-based modeling community has been wondering how to model discrete data for some time, and this paper provides an answer to this question. That being said, the main technical contribution amounts to using a modulo operator (Eq. 5) and handling backpropagation through an argmax operator (Eq. 6) on top of the existing techniques of MAF and Real NVP. I view this simplicity as a benefit of the approach, but some may view this a simple extension of existing techniques. Quality: The technical and experimental aspects of the paper are well-executed. The authors provide multiple experiments to demonstrate autoregressive and bipartite flows. Within these experiments, various hyper-parameter settings are reported to gain a better intuition for the performance of the models. Generation time is reported, helping to demonstrate the benefits of the models. For the most part, the technical ideas are fully developed and explored. Clarity: The presentation of the approach is incredibly clear. Examples are given during the presentation, which help the reader gain intuition about when the approach is useful. The diagram in Figure 1 is helpful for unfamiliar readers. For the most part, the experiments section is also clear. Some details of the models and training set-up are unclear, particularly in the toy examples from sections 4.1 - 4.3. Additional details in the supplementary material would help to clear up confusion. Significance: Although the introduction of discrete flows is a significant contribution, the paper currently feels like more of a proof-of-concept, rather than a competitive new approach. Demonstrations of new techniques are helpful, as other researchers will undoubtedly extend this technique to new settings. But additional experiments would help to complete this paper and broaden its impact. Many flow-based models have been applied to images, and it seems like discrete image datasets, e.g. binarized MNIST or Caltech-101 silhouettes, would be a natural testbed. In fact, RGB images are already naturally discrete. Likewise, with recent interest in discrete latent variable models, e.g. VQ-VAE, applying inverse autoregressive flows for variational inference would be another natural choice. --- Updates: Assuming the authors provide additional details on experiments in the supplementary, then I will be happy with this aspect. I'm perplexed as to why the authors seem resistant to running experiments on a simple binary image dataset, e.g. binarized MNIST or Caltech-101 Silhouettes. With binary data, there wouldn't be any issues with the ordinality of the pixels. And these datasets are small enough that getting results should take a matter of hours or less. This just seems like an obvious experiment to try to see how discrete flows compare with other families of generative models. It would also help to broaden the appeal of the paper to a wider audience.

Reviewer 3



The paper highlights that, despite not looking very obvious at first, normalising flows are in principle available for discrete data as well. The key is to design invertible transformation between discrete spaces. We can think of such a bijection, as a relabelling of the discrete sample space, and there's no need for computation of determinant Jacobians. They also show that it's not difficult to design parametric invertible transformations. They show an example with XOR and a generalisation thereof based on mod K. The real difficulty--and the paper could be more explicit here--is how to estimate the parameters of such discrete transformations. Their parameters are themselves discrete, thus if you have a NN predict them (for maximum flexibility), this network would require a nondifferentiable output activation. The approach taken at this point is to ignore the problem, and employ the straight-through estimator (which the authors argue work well for problems where K is not too large). The authors demonstrate the technique is effective in controlled artificial tasks as well as in char-level density estimation for text (PennTreebank and text8) showing both improved likelihoods and fast generation. The paper is clearly original, and I imagine it will be of great significance (it brings two interests together, namely, flexible flow-based density estimation, and modelling discrete data). The paper is mostly quite clear, I only have a few remarks in the next box.

[Author Response · NeurIPS 2019]

Thanks for all the comments! We answer major comments from each reviewer below; we'll fix the minor ones.

**REVIEWER 1:** **"This paper ranks high in novelty...The experimental results are strong, especially on Text modelling. Moreover, the proposed method significantly computationally more efficient compared to competing approaches...The paper is very well written and easy to understand."** Thanks for the high praise!

**"1. The authors should consider a wider array of distributions to more convincingly demonstrate the capabilities of the proposed flow layers."** We include many synthetic discrete problems in the Experiments section: full rank discrete distributions, addition, and Potts models. We decided against including more continuous/ordinal data problems as our flows don't take advantage of ordinal structure in the Categorical states, and there are many practical problems without this structure.

**"2. Some important details are unclear. E.g. what is the base distribution for sampling? Is it the factorised marginal distribution? If it is how is it estimated in high dimensions (given that the number of data samples needed for an accurate estimate would grow exponentially)?"** We use factorized Categorical base distributions for bipartite flow models and correlated Categorical base distributions parameterized by an autoregressive neural network for autoregressive flow models. Note that the number of parameters for the base distribution is far less than the total number of possible states (e.g., $K \cdot D$ for factorized Categorical instead of $K^D$).

**REVIEWER 2:** **"Originality: This paper is the first demonstration of flow-based models to discrete data. As such, the work is fairly novel....That being said, the main technical contribution amounts to...on top of the existing techniques. I view this simplicity as a benefit of the approach, but some may view this a simple extension of existing techniques."** Thanks for the high praise! We agree about simplicity being a benefit. We explicitly designed discrete flows to be natural extensions of the continuous case.

**"Quality: The technical and experimental aspects of the paper are well-executed. Clarity: The presentation of the approach is incredibly clear."** Thanks!

**"For the most part, the experiments section is also clear. Some details of the models and training set-up are un-clear, particularly in the toy examples from sections 4.1 - 4.3. Additional details in the supplementary material would help to clear up confusion."** Great idea. We'll clean this up .

**Adding discrete image datasets.** Demonstrating discrete flows on image data is a good idea. So far, we focused on text applications to show a domain normalizing flows hadn't yet been applied to. It's a ripe area given that both flexible modeling with bidirectionality and nonautoregressive generation are of huge interest for text. We'd love to explore images in future work, in particular pushing on the ordinality of pixel intensities to better handle data quantization. Hoogeboom et al. (2019) provide excellent complementary work in that direction.

**"In Section 3.1, 3.2., or in the supplementary, it would be helpful to have an expanded discussion around when discrete flows are invertible and what difficulties there are in ensuring this aspect. This discussion could also include the invertible discrete functions alluded to in Section 5."** In the revision, we'll add a section about the expressivity of discrete flows, what the set of transformations are (permutation-based), and designing parameterized invertible discrete functions.

**REVIEWER 3:** **"The approach taken at this point is to ignore the problem, and employ the straight-through estimator (which the authors argue work well for problems where K is not too large)."** The straight-through estimator is effective and commonly used in many discrete optimization problems to compute gradients through the non-differentiable argmax operation. In future work we hope to investigate whether other gradient approximations (including non-gradient based optimization) improve performance.

**"The paper is clearly original, and I imagine it will be of great significance (it brings two interests together, namely, flexible flow-based density estimation, and modelling discrete data)."** Thanks!

**"Since you count on backpropagation via straight-through estimator (STE), the derivative of mod K becomes relevant (as it will be necessary for chain rule to update the parameters of the NN that predict flow parameters). The best I could gather is that it's probably 1 everywhere except when (u + sigma x) is exactly divisible by K. Is that correct?"** You're correct that the operations are non-differentiable. The goal with continuous relaxations is not to compute the true gradients, but rather to provide a useful signal for improving the loss. We'll add these nuances to the paper.

**"You mention STE works well if K is not too large, but is that all?"** That's a great question. Depth (number of flows) affects gradients, since the bias explicitly accumulates as each flow uses an approximation. This is also the case for dimensionality (sequence length). We haven't found complexity of the networks parameterizing the flow to make a difference, but this requires further investigate. Note an additional complication follows your above point: "true gradients" are not well-defined. So instead of examining gradient bias, bias may be formalized by comparing, for example, the minima from the discrete loss to the minima from the relaxed loss.

[Meta-Review · NeurIPS 2019]

The authors develop autoregressive and bipartite discrete formulations of discrete flows. The reviewers felt the paper represents significant conceptual advances. However, there were some remaining concerns after the rebuttal period about the experiments. For example: "I'm perplexed as to why the authors seem resistant to running experiments on a simple binary image dataset, e.g. binarized MNIST or Caltech-101 Silhouettes. With binary data, there wouldn't be any issues with the ordinality of the pixels. And these datasets are small enough that getting results should take a matter of hours or less. This just seems like an obvious experiment to try to see how discrete flows compare with other families of generative models. It would also help to broaden the appeal of the paper to a wider audience." Please carefully account for (updated) reviewer comments in your revisions.